# Impact of crowdfunding, entrepreneurial finance and varieties of entrepreneurial ecosystems after COVID pandemic for rural women

Zhikang Lyu[1]*, Natasha Murtaza[2]

1 Guangdong University of Finance, Guangzhou, China, 2 Institute of Agricultural and Resource Economics, Faculty of Social Sciences, University of Agriculture Faisalabad, Pakistan

* zhikanglyu81@gmail.com

## Abstract

In this paper, we propose an approach to estimate the role of entrepreneurial finance and social capital distribution in crowdfunding (CF) and varieties of entrepreneurial ecosystems after the COVID pandemic for rural women in Pakistan. Primary data were collected from 1,004 rural women. Using SmartPLS, hypotheses were analyzed through partial least squares structural equation modeling. The mainstream tests used for checking the discriminant validity are the Fornell and Larcker Criteria, Heterotrait–Monotrait ratio, Average Variance Extracted, Variance Inflation Factor (VIF), and the factor loadings of each item for each variable. Similarly, the reliability of the data was analyzed through Cronbach's alpha and composite reliability. The results showed that crowdfunding, social capital distribution, and the variety of an entrepreneurial ecosystem could play a vital role in the entrepreneurial intentions of rural women in Pakistan. Moreover, the distribution of social capital plays a positive role in the variety of the entrepreneurial ecosystem. The study also provides a benchmark pathway for social change for rural women in Pakistan following the COVID-19 pandemic, revealing how various entrepreneurial ecosystems can assist them in achieving their business goals by providing a clear roadmap for crowdfunding (CF).

## 1. Introduction

Women's empowerment has emerged as one of the world's significant priorities for long-term development. This aspect is being examined even in developed economies, as women in developed nations face many kinds of discrimination [1]. Similarly, women's economic empowerment has been identified as a critical predictor of long-term prosperity and women's well-being [2]. Pakistan is a developing nation that has historically been largely rural yet has the fifth-highest population in the world. The National Institute of Population Studies estimates that Pakistan will have 224.78 million people in 2021, with 82.83 million living in urban areas, 141.96 million in rural

**Data availability statement:** All data is available on the following link publicly: https://www.openicpsr.org/openicpsr/project/241029/version/V1/view.

**Funding:** The author(s) received no specific funding for this work.

**Competing interests:** The authors have declared that no competing interests exist.

areas of the country, and a population density of 282 people per square kilometer [3]. Vision 2030 strongly emphasizes gender equality via empowering women [4]. Similarly, Pakistan is committed to the Beijing Platform for Action, The International Labor Organization conventions, the Child Rights Conventions, and the Committee on the Elimination of Discrimination Against Women, all of which aim to secure women's appropriate role in society [5]. There has been a major innovation in recent years, which may be attributed to increased strategic investments made to enhance the livelihoods and well-being of women and girls in Pakistan.

Women comprise 48.4% of Pakistan's population [6]. Women's equality, empowerment, involvement, and representation in all phases of life are essential for Pakistan's sustainable socioeconomic, political, and cultural growth [7]. Despite this, women's standing is subpar. In Pakistan, women deal with various issues, including domestic abuse, acid throwing, humiliation, and honor killings [8]. In Pakistan, gender inequality is widespread and might restrict the country's socioeconomic development [9]. Even though there are differences in their implementation based on rural–urban location, geographical region, and status, norms still exist that limit women's participation in social activities [10]. Although it has increased gradually, their participation in official political processes remains below what would be expected, given their demographic share. Women's influence on decision-making is very minimal. Both new and existing enterprises are started and nurtured by rural women [11]. Due to discrimination, women have frequently been deprived of opportunities. The poverty rate in Pakistan has continued to rise due in large part to the limited economic progress of women [12]. Because of this, women in remote parts of villages become vulnerable, necessitating a focus on crowdfunding and various entrepreneurial ecosystems that are affordable to the underprivileged [13]. More creative funding methods are required to increase the economic growth of rural women.

Entrepreneurs in need of funding who have creative ideas generally turn to banks, venture capitalists, and other sources for their funding [14]. The funding for Pakistani women under the Kamyab Jawan Program (KJP) is 25 Billion Rupees (approximately 1 Rupee = 0.03 USD) [15]. The Youth Entrepreneurship Scheme (YES) of the KJP has certified Pakistani female entrepreneurs, and the government has so far distributed around 1 billion Rupees to them [16]. There is a significant argument about whether entrepreneurship benefits rural Pakistani women, even though women's entrepreneurship has been connected to empowerment. In previous studies (given in Table 1 and Section 1.1) on rural women's development, the quantity of loans they may obtain from nonprofit organizations and general livelihood programs has been examined as a factor.

### 1.1. Crowdfunding

Crowdfunding (CF) is a unique strategy for encouraging and funding the growth of startups, individuals, and ecosystems. Crowdfunding (CF) has also become popular in developing economies as a distinctive approach to funding entrepreneurial projects particularly to individuals and ecosystems that have difficulty with conventional funding avenues [17]. Recent studies indicate that CF is under-researched in South Asia,

**Table 1. Related literature review on CF.**

| Sr. No | Authors | Keywords | Database | Research Design | Key Findings |
|--------|---------|----------|----------|-----------------|--------------|
| 1. | (André et al., 2017) | Crowdfunding and Reward based crowdfunding | Web of Science, Scopus, Google Scholar | Quantitative | Crowdfunding platforms contribution in Entrepreneurship |
| 2. | (Butticè et al., 2017) | Serial Crowdfunding, Social Capital, and Project Success | Science Direct | Quantitative | Serial crowdfunders utilize their social connections with individuals who supported their prior campaigns. |
| 3. | (Gleasure et al., 2017) | Crowdfunding, Socio materiality and unbond | Google Scholar and Science Direct | Qualitative | Unbound assists authors in obtaining financing by using crowdsourcing techniques. |
| 4. | (Johnson et al., 2018) | Crowdfunder judgments, implicit bias and Crowdfunding | Science Direct | Quantitative | Women fundraisers are more likely to be successful because they are more trustworthy than men. |
| 5. | (Nielsen, 2018) | Crowdfunding and Organization | Web of Science, Scopus, Google Scholar and Science Direct | Conceptual | Crowdfunding works as a network that leads to community projects. |
| 6. | (Butticè & Noonan, 2019) | Crowdfunding, Social Capital & Social Obligation | Google Scholar and Science Direct | Quantitative | Quality of the product & commercialization following a reward-based crowdsourcing effort. |
| 7. | (Z. Wang et al., 2019) | Financing Reputation and Social Network of funding | Scopus, Google Scholar and Science Direct | Quantitative | Campaign success is related with effective communication. |
| 8. | (Strohmaier et al., 2019) | Trust, Crowdfunding and Institutional Mechanism | Google Scholar, Science Direct, EBSCO, Scopus, Web of Science | Quantitative | Trust improves attitudes about initiatives, which increases financial intentions. |
| 9. | (Dejean, 2020) | Geography and Crowdfunding | Web of Science | Quantitative | Increased market flows result from stronger social networks between regions. |
| 10. | (Figueroa-Armijos & Berns, 2022) | Crowdfunding and Female and Rural Entrepreneur | Google Scholar, Science Direct, and Web of Science | Qunatitative | The relevance of vulnerability in effectively obtaining donations in the context of socialization crowdfunding. |

given its potential to help develop it in a sustainable way and empower marginalized populations [13]. An example is, the power of CF in fostering social capital in developing economies is proven by platforms like Kiva.org that help low-income entrepreneurs [18]. This research can fill the literature gap because it will examine how CF can facilitate entrepreneurial intentions among women in rural Pakistan, especially after the COVID-19. According to some researchers, CF, which has a history that extends back to the 17th century, is a kind of crowdsourcing. CF has been practiced since the 1800s. Then, in 2000, Artist Share, a website where artists could ask their followers for donations to create digital recordings, was founded in the United States. Using CF, for instance, may help ecosystems find and include new customers, clients, and individual and professional investors. In the alternate finance sector, CF has recently grown in popularity as a way of raising money and as a feasible and growing alternative method of supporting various projects. The prospect of CF in the developing world was published in 2013, and the World Bank estimated there was a market opportunity worth up to USD 5 Billion in South Asia [19]. CF is a method of raising funds for a business or a philanthropic or personal enterprise by seeking small sums of money from many individuals [20]. Whereas the internet has made the process much more accessible, the concept is not new and can even be traced back to the 1885 fundraising for the Statue of Liberty [21].

As a substitute for bank loans and equity capital, CF has the potential to expand capital availability for emerging commercial and social entrepreneurs and those seeking investment [17]. CF portals provide a digital platform for relevant stakeholders to connect and generate possibilities, primarily for entrepreneurs and active investors [22]. Following the reasoning above, forums provide community newsletters and communications tools for all types of campaigns, allowing entrepreneurs to communicate with and recruit fellow entrepreneurs [23].

 

There have been just a few studies on CF in South Asia, but improvements are definitely needed. Studies on CF in India looked at drivers of economic growth, risk considerations, regulatory constraints, and the development of business models [24]. Similar studies in the Bangladesh area focus on the business strategy behind CF and the awareness and incentives behind it [25]. Few CF platforms existed in India before 2014, and the public assumed that the success of a CF campaign would be determined by family, friends, or venture capital firms [26]. On the other hand, the Bangladeshi CF platforms that are now available were not completely operational in 2019 [27]. This research advances our understanding of CF from the perspective of the development of rural women in Pakistan.

Although there is not a wide range of literature on CF, there is rising interest in the many ways that value is traded across actors. According to [28], "an initiative done to gather money for a new project presented by someone, by accumulating small to medium size funds from a number of other people." Rural women may use CF to achieve social value through entrepreneurial activities [18]. Several studies on CF were based on Kiva.org, a US-based platform that provides low-income business owners with small-dollar loans [29]. However, there is no concrete data on the effects of CF, particularly regarding rural women borrowers and development. By analyzing the elements that contributed to rural Pakistani women's success, this research intends to shed some light on this. Table 1 shows related literature reviews on crowdfunding.

## 1.2. Varieties of entrepreneurial ecosystems

The study of entrepreneurial ecosystems (EE) has gained popularity among academics and decision-makers [30]. As a result of these debates, many concepts and principles with a variety of connected pieces have been developed. In the 1980s and 1990s, when entrepreneurship research shifted from an individualistic perspective to a more comprehensive perspective that emphasized the participation of economic, political, and social variables in the entrepreneurship process, the fundamentals of the idea of an EE emerged [31]. Although there is no one definition of an EE, it has been interpreted in various ways by academics, researchers, and professionals. Many comparisons have also been made, including business, industrial, innovation, and entrepreneurial ecosystems [32].

The term "ecosystem" has its roots in nature and has since been adopted by fields including economics, business, and human sciences [33]. Scholars characterize EE varieties as integrated environments that promote successful entrepreneurial opportunities for rural women in Pakistan [34]. The most common definition of an EE is a locally constrained network of organizations and people that supports entrepreneurs in recognizing and seizing market opportunities. However, having the prospect in the first place is necessary to take advantage of an entrepreneurial opportunity. Entrepreneurial ecosystems (EE) have become an important concept in the interpretation of the interdependent variables that affect the entrepreneurial activity. According to a research by [35] to develop entrepreneurial intentions, there must be a favorable ecosystem, especially in rural regions with limited resources. An overall EE encompasses a range of factors like access to finance, enabling policies, networks, and social capital, which play a significant role in making entrepreneurial ventures successful [34]. In South Asia, entrepreneurial ecosystems are still on their infancy, and networking barriers, as well as access to resources, are severe. Nonetheless, entrepreneurial ecosystems can also foster the development of rural women entrepreneurship in countries such as Pakistan, as pointed out by [34], since these ecosystems can introduce them to crucial resources and mentors. This paper examines the impact of various EEs on the entrepreneurial intentions of rural women and how these ecosystems can be used to enable them to access crowdfunding. A sustainable EE is "an integrated and collaborative set of stakeholders offering sustainability-oriented assistance to entrepreneurs in terms of developing entrepreneurial ventures that effectively address the economic, environmental, as well as social dimensions of sustainability and so contribute to the transformation to a sustainable regional economy." Table 2 shows related literature reviews on Varieties of Entrepreneurial Ecosystems.

## 1.3. Conceptual framework

Many researchers claim that women-run enterprises create jobs, earn substantial returns on investments, and contribute to poverty alleviation [17]. This conceptual framework (Fig 1) outlines the role of CF and varieties of EEs for rural Pakistani women and their overall growth considering the previous theoretical and empirical findings. CF and varieties of EEs play a positive and significant role in the entrepreneurial

**Table 2. Related literature review on varieties of EEs.**

| Sr. No | Authors | Focus | Database | Keywords | Key Findings |
|---|---|---|---|---|---|
| 1. | (Alvedalen & Boschma, 2017) | Varieties of Entrepreneurial Ecosystems and Entrepreneurship | Scopus and Web of Science | Entrepreneurial Ecosystems, Entrepreneurship, Networks, Clusters & Entrepreneurial Systems | Varieties of entrepreneurial ecosystem (EE) has received a lot of attention, particularly in government circles. |
| 2. | (Belitski & Heron, 2017) | Expanding the Entrepreneurship Education Ecosystems | Social Sciences | Varieties of Entrepreneurial Ecosystems, university education, innovation, learning | Entrepreneurship ecosystems in education have emerged as the most essential and effective tool for business community. |
| 3. | (Zhang & Guan, 2017) | Entrepreneurial Ecosystems, and Entrepreneurs | Web of Science | Ecosystem, Innovation & Entrepreneur value | Trends and features in varieties of entrepreneurial ecosystem. |
| 4. | (Dedehayir et al., 2018) | Innovation in Ecosystem | Google Scholar & Web of Science | Ecosystem genesis, Innovation ecosystems, And Ecosystem roles | To strengthen our knowledge of the origins of innovation ecosystems. |
| 5. | (Malecki, 2018) | Entrepreneurial Ecosystems, Entrepreneurs and Entrepreneurship | Google Scholar, Web of Science, and Scopus | Entrepreneur, Startup ecosystem, Entrepreneur, Infrastructure and Entrepreneur System | Concept of usage of the word entrepreneurial ecosystems |
| 6. | (Cavallo et al., 2019) | Emerging Entrepreneurial Ecosystems | Web of Science and Scopus | Potential Entrepreneur, Ecosystem, and Entrepreneurial System | The basic components that comprise an entrepreneurial ecosystem |
| 7. | (Robertson et al., 2020) | Growing varieties of Entrepreneurial ecosystems | Web of Science, Scopus, ABCD and Google Scholar | Varieties of Entrepreneurial Ecosystems | Review of the growing entrepreneurial ecosystems in their many aspects |
| 8. | (Velt et al., 2020) | Entrepreneurial Ecosystems | Pro Quest, EBSCO, Scopus, and Web of Science | Entrepreneur, Ecosystem, and Entrepreneur Infrastructure | Research on Entrepreneurial Ecosystems |
| 9. | (Wurth et al., 2021) | Entrepreneurial Ecosystem and Research Program | Web of Science CoreCollection, Scopus, and Google Scholar | Entrepreneur, Ecosystem, Entrepreneurship, Research Program | Examines the empirical observation of the underlying processes and takes inventory of recent developments in the study of entrepreneurial ecosystems. |
| 10. | (Fernandes & Ferreira, 2022) | Entrepreneurial ecosystems and startups | Science Direct and Scopus | Entrepreneur, Ecosystem, Startup ecosystem and Networks | Conceptual Framework of Entrepreneurial ecosystems and networks |

intentions of rural women in Pakistan. Using the conceptual framework and social capital theory, we investigated the role of CF, social capital distribution and varieties of EEs from 1,004 rural Pakistani women through the snowball sampling technique. This conceptual framework also shows the relationship between entrepreneurial finance and social capital distribution among rural women in Pakistan. In this conceptual framework, the impact of entrepreneurial financing and social capital distribution on CF and other EEs, as well as their combined influence on rural Pakistani women's entrepreneurial intents, has been evaluated. The study has also examined the impact of various EEs on rural Pakistani women's access to CF. In a nutshell, given the limited amount of research on the topic in Pakistan and the country as a whole, this conceptual framework assists in evaluating these factors' overall influence on the challenges rural women face in Pakistan.

## 1.4. CF has a positive impact on entrepreneurial intentions

CF is a new phenomenon in Pakistan that has attracted the curiosity of professionals and academics due to its appeal as an alternative funding source [36]. CF's development has been tremendous, and new opportunities have developed. CF is a brand-new method of raising capital and promoting new products and ideas [37]. This study looks at the variables that affect rural Pakistani women's intentions to participate in CF. In consideration of previous research, we proposed this hypothesis:

**H1:** *CF positively impacts the entrepreneurial intentions of rural women in Pakistan.*

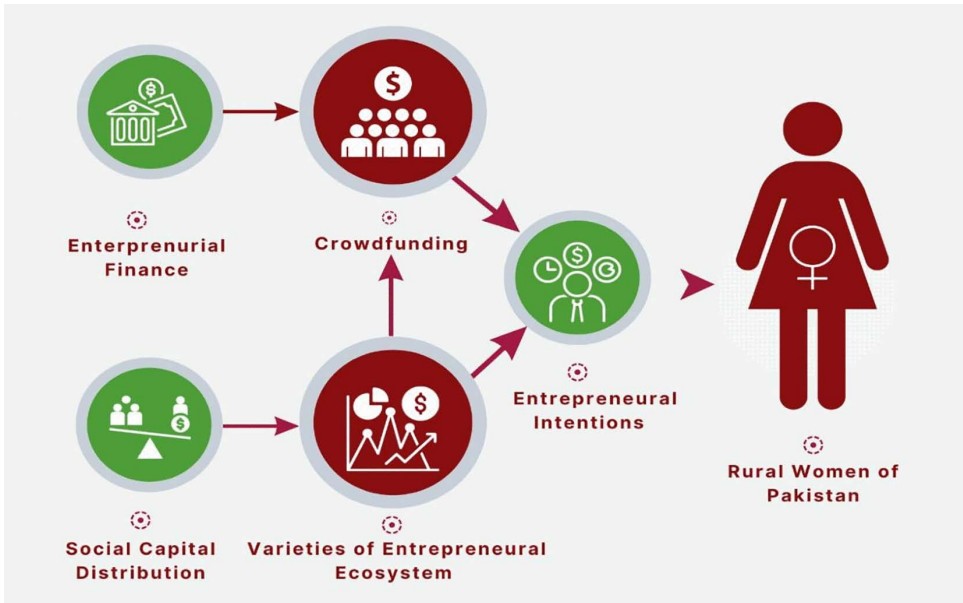

**Fig 1. Conceptual framework on role of CF and varieties of EEs for rural women in Pakistan.**

## 1.5. Varieties of EEs positively impact entrepreneurial intentions

Researchers, policymakers, and entrepreneurs now devote more attention to EEs, considering that they are essential for building sustainable economies based on entrepreneurship and innovation during the past fifteen years [38]. Despite the increasing academic research, [39] claims that most EE literature is still practice-focused. EE research is in development, with no coherent theory or study aim. The varieties of EEs as integrated environments promote successful entrepreneurial intentions and opportunities for the rural women of Pakistan. So, to analyze the impact of varieties of EEs on entrepreneurial intention, we hypothesized that:

**H2:** *Varieties of EEs positively impact the entrepreneurial intentions of rural women in Pakistan.*

## 1.6. Entrepreneurial finance positively impacts CF

Entrepreneurial finance is essential for benefiting from entrepreneurial activities. More people are working in the informal sector in almost every developing nation. There are several informal sectors, including small- and medium-sized businesses (SMEs) that are individual-owned startups. Entrepreneurial finance (EF) is relevant to assist the entrepreneurs, especially in the developing economies that lack the use of traditional financial sources [19]. Studies have indicated to the effect that EF, both formal and informal financial sources, have extensive influence on the growth and sustainability of new ventures [40]. Among the rural women entrepreneurs, entrepreneurial finance is a very important tool in tackling financial barriers and facilitating the establishment of business [41]. Research has shown that rural South Asian women have unique barriers in the utilization of financial resources, which makes EF mechanisms especially important in supporting their entrepreneurial intentions [29]. Governmental programs like Kamyab Jawan Program (KJP) in Pakistan have been aimed to empower young women entrepreneurs, which highlights the significance of EF in helping the gender gap in entrepreneurial finance [15]. This industry employs many women, and such an unorganized industry is essential for capital savings and employment. For many years, both entrepreneurship researchers and economists have been interested in the topic of entrepreneurial financing. This interest has grown out of the necessity to understand and address problems faced by entrepreneurs and their

significance in the growth of entrepreneurial businesses. The financial school of thinking often sees entrepreneurial finance as a fundamental pillar since it is required to launch a new firm or enterprise [22]. Entrepreneurial finances in entrepreneurship generally refer to the entrepreneurs' accessibility to and number of adequate options. CF is a relatively new and rapidly increasing phenomenon in entrepreneurial finance, allowing entrepreneurs to request financing from a potentially large number of investors. We proposed this hypothesis to check the positive impact of entrepreneurial finance on CF:

**H3:** *Entrepreneurial finance positively impacts CF for rural women in Pakistan.*

## 1.7. EEs positively impact CF

EEs are frequently described as spatially constrained networks of organizations and people that support entrepreneurs in identifying and exploiting market opportunities [34]. Cultural, psychological, political, and economic factors all work together in EEs to affect the emergence and development of new businesses [42]. However, getting the chance in the first instance is essential for just being able to take advantage of an entrepreneurial venture. A comprehensive review of the CF phenomena demands an EE approach. CF's contribution to entrepreneurship, job creation, and economic growth has proliferated throughout Europe. By developing an innovative regulatory framework for CF that takes advantage of the speed of technology and new innovation strategies, the opportunity for both startups and SMEs to access the capital they need to build and operate while better protecting investors, European policymakers can encourage the development of an allowing ecosystem for entrepreneurship. So, that is why we proposed this hypothesis:

**H4:** *EEs positively impact CF for rural women in Pakistan.*

## 1.8. Social capital distribution for rural women in EEs in Pakistan

Social capital distribution and network hurdles for female business owners result from distorted expectations, stereotypes, and basic ideas underlying capitalists' unconscious and conscious search parameters [34]. As a result, women are frequently refused permission to high-level networks in politics and business controlled by men in Pakistan's rural areas. Such patriarchal arrangements were found to be particularly common in developing nations, such as Bulgaria, Moldova, and even Ukraine. Such restrictions are less noticeable in more developed countries like the United States, where government organizations offer a wide range of assistance mechanisms based on the sector, entrepreneurial initiative type, or entrepreneur-to-be's socioeconomic standing. Understanding social stratifications in an entrepreneurial environment requires considering various social capital, a fundamental theoretical approach [33]. This understanding is critical in high-growth EEs because female entrepreneurs face challenges accessing network resources, such as advisers, investors, and mentors. There are still many unaddressed concerns about EEs, even though research on female entrepreneurs' social capital has grown substantially over the years. Our work conceptualizes this idea to respond to some of these queries. Researchers discovered that despite getting loans with comparable terms, female company owners felt that financing agents treated rural Pakistani women with disproportionate disrespect [16]. As a result, previous academic research reveals that social capital distribution has distinct effects on men and women at different phases of the entrepreneurial process and within the EE. In consideration of this, we suggest the following hypothesis:

**H5:** *Social capital distribution for rural women in EEs in Pakistan.*

## 2. Materials and methods

The present study has measured the effect of entrepreneurial finance and social capital distribution on CF and varieties of EEs and their collective role in the entrepreneurial intentions of rural Pakistani women. The study has further estimated the role of varieties of EEs on CF for rural women in Pakistan for the first time. The study has developed five hypotheses based on the literature reviewed on CF, EEs, and entrepreneurship. A quantitative study design was established to ensure biases, such as social desirability and researcher bias, did not affect the study results. A proper overview of the research design is shown in Fig 2.

S1 Table in supplementary file illustrates the questionnaire items of each latent construct. These were the adoptions of the scales in the literature. As an illustration, the items on the 'Entrepreneurial Finance' construct were taken out of the work of [43] the items on the construct of Social Capital Distribution were created out of the previous works in this field. All the items are listed below to get content validity.

The questionnaires were self-administered and the researchers gave advice to the participants where necessary to facilitate comprehension, especially to those with low educational attainment. The response rate was 1004 filled questionnaires out of 1050 mailed, which constituted 95 percent response rate. A pilot test was carried out to test the question clarity and the general questionnaire structure before final deployment using 100 participants to determine the clarity of questions and the overall questionnaire form. According to the responses of the pilot test, some slight modifications were introduced to make the questions culturally appropriate and comprehensible by the respondents.

Both theoretical and empirical factors informed the choice of items in each construct. The selection of items was done according to the relevancy to the conceptual definition of the construct and their capacity to reflect the essence of

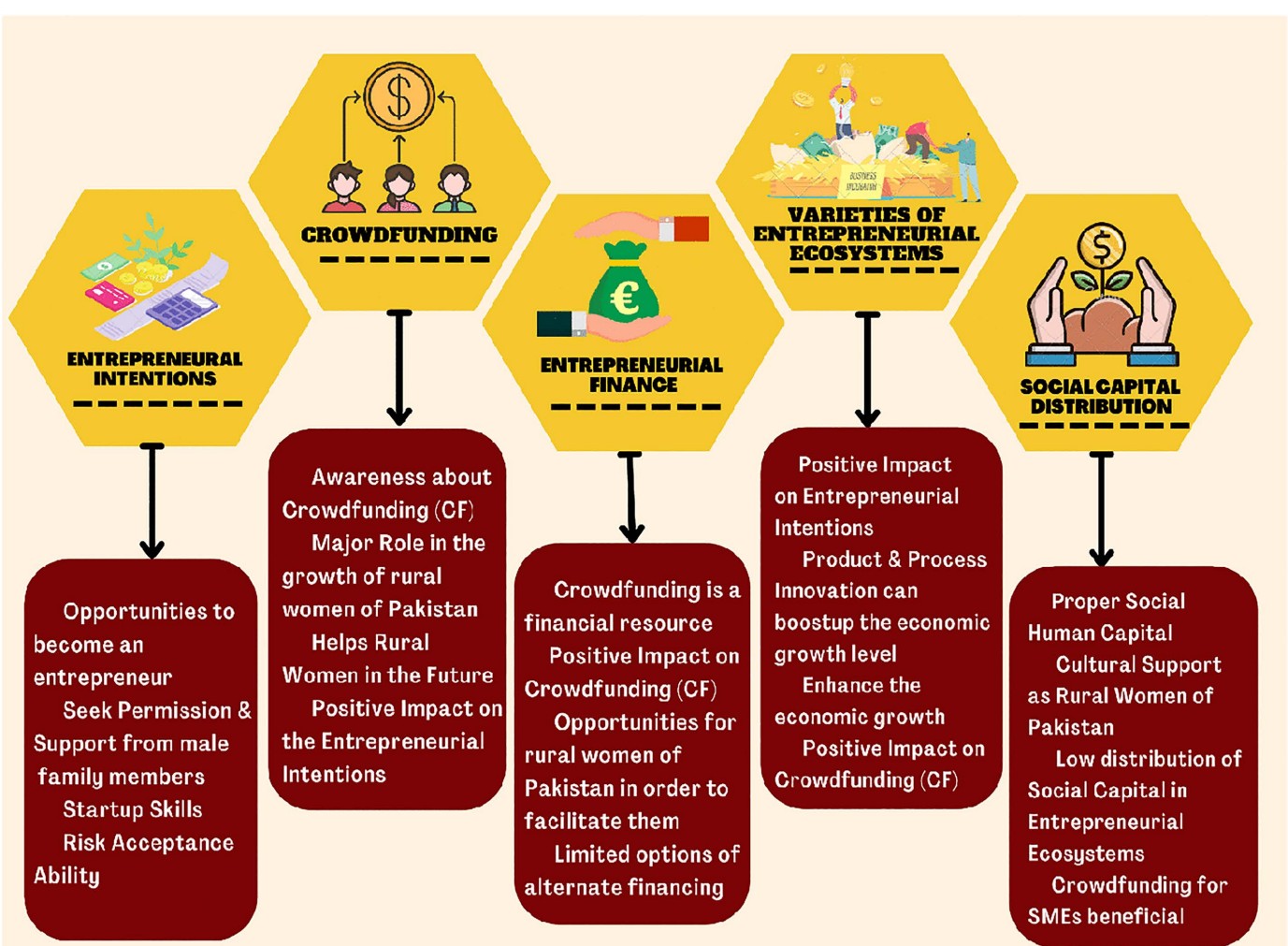

**Fig 2. Research design of CF and varieties of EEs for rural women in Pakistan.**

the theoretical model. In the case of Entrepreneurial Finance, items were selected to represent both the accessibility of finance and the role of financial networks in attending to the rural women entrepreneurs. The pilot test (100 participants) was conducted to examine the clarity and relevance of each item and make sure that this is construct valid.

These findings suggest that key relationships have a beta of between 0.35 to 0.60 with the biggest effect size being between the relationship between Entrepreneurial Finance and Crowdfunding (beta = 0.60), indicating a significant practical relevance. Conversely, the correlation between the Social Capital Distribution and Entrepreneurial Intentions had a medium impact (beta = 0.45) which means that there is a large but not significant influence. These results indicate that although each of the constructs is significant in influencing entrepreneurial intentions, the construct that carries the greatest practical significance on crowdfunding success among women in rural Pakistan is the Entrepreneurial Finance.

To further enhance the analysis, we will develop the mediation and moderation analysis to investigate the possible indirect effects and conditional associations between the important variables. As an illustration, we hypothesize that the moderating role of 'Social Capital Distribution' could exist between the relationship of 'Entrepreneurial Finance' with 'Crowdfunding' success, and that the effect is stronger in networks featuring high social capital. This further discussion will be useful in explaining how these factors affect the entrepreneurial intentions of rural women. To estimate it model, the stability and significance of the indirect effects in the structural model were tested by bootstrapping with 5,000 subsamples. Each path coefficient was also calculated and the results showed that all the main relationships were significant at the 95% confidence level. Such bootstrapping process adds more strength and reproducibility of the results because the possible sampling variability is considered.

The entire set of questions in this study together with items that represent each of the latent constructs is provided in Appendix X. The items were also chosen well to fit the theoretical definitions of the constructs using available scales in literature. Expert reviews and pre-testing were used to measure construct validity. A test sample of 100 participants was used to test the clarity and relevance of the final items before implementing them in the main study.

## 2.1. Study area

Pakistan is a data-scarce country, and we spent more than six months among rural women in Pakistan. Additional respondent remarks are carefully recorded throughout the field investigation, and the researcher reread these minutes daily. These factors further increased the reliability of the primary data, which helped the researcher produce better findings. The anonymity of the respondents was also ensured. Statistical software aided the analysis of the data collected through a survey to measure the developed hypotheses.

**2.1.1. Data collection and sampling technique.** In this study, the Snowball Sampling Technique for sample selection was adapted to collect data for the research. The Snowball Sampling Technique helps identify and access the participants through their social circle [44,45]. In this technique, the first person accessed is the source for connecting the next potential participant. More than 1,000 participants were contacted this way and completed the questionnaire. Larger sample sizes assist researchers in identifying the outliers in data and give narrower margins of error. The surveys were self-administered as the data were collected in batches according to the participants' availability. This study's respondents were women from rural areas in Pakistan who tend to start their businesses. Each variable consisted of four items. Screening of the questionnaires indicated that 1,004 out of 1,050 questionnaires could be used for data analysis. The respondents were made to feel comfortable while participating in the survey to ensure free and easy responses. Data survey permission was acquired from the study area and ethocs board of Directorate of Institute of Agricultural and Resource Economics, University of Agriculture Faisalabad, Pakistan. The reason why snowball sampling has been adopted is because it is challenging to access the target population (rural women with entrepreneurial intention) using traditional sampling techniques. In rural Pakistan, where some groups may be inaccessible in some instances because of social, economic and geographical constraints, snowball sampling would provide more efficient access to participants using personal networks. This technique has effectively been applied in related studies, especially in situations where

the focus is on a hard-to-reach population [34]. Although snowball sampling is prone to bias, since respondents are usually members of the same social or economic status, these risks were prevented. We tried to make our sampling more balanced by starting with a variety of initial participants in different rural areas and making sure that they referred to women of different backgrounds and places.

Note: A statement to confirm that all methods were carried out in accordance with relevant guidelines and regulations. All necessary protocols were followed and there were no physical tests. Only perceptions based data were recorded after respondents' consent. Oral consent was taken as no sensitive or personal data were recorded. Data were collected from September 2024 to November 2024.

## 2.2. Questionnaire design

The questionnaire distributed among the participants was designed on a five-point Likert scale. The response categories ranged from 1 to 5, with 1 being a strong disagreement and 5 being a strong agreement. The questions in the questionnaire addressed each variable (i.e., entrepreneurial finance, social capital distribution, CF, EE varieties, and entrepreneurial intention).

**2.2.1. Independent variables.** Social capital distribution and entrepreneurial finance have different circumstances that require capital or financial support [18]. Finance is required to increase investments and create liquidity. Although businesses require money, it might be challenging to get loans. Entrepreneurial financing is required to benefit from the advantages of entrepreneurial activities. Entrepreneurial opportunities are inherent in the economy, but entrepreneurial development and implementation are unique processes requiring firm-level activity.

**2.2.2. Dependent variables.** CF is a rapidly expanding trend in entrepreneurial financing that allows individuals to seek funds from many possible investors [36]. The current study evaluated the impact of entrepreneurial financing and social capital distribution on CF and other EEs and their combined influence on rural Pakistani women's entrepreneurial intentions. CF and varieties of EEs play a positive and significant role in the entrepreneurial intentions of rural women in Pakistan.

**2.2.3. Pre-testing approach.** A pre-testing approach was utilized in this research to achieve the goal of accuracy and clarity. One hundred respondents were chosen to verify the data's accuracy, and the final questionnaire was modified suitably after receiving the respondents' final responses.

## 2.3. Statistical tools

Structural equation modeling (SEM) analysis determines the data's actual behavior. The statistical analysis followed the partial least squares method using Smart PLS software, which helps in analyzing the data through path modeling in a short time. This method is advantageous for the researchers since it is a variance-based, prediction-oriented software that explicitly estimates the latent variable scores [46]. Researchers also prefer this software over others because of this its ability to handle small to large sample sizes with accurate results. The power analysis in this software is generally based on the portion of the model with large samples. The recommended small sample size is optimally said to be between 30–100 [47]. In this program, one variable acts as the dependent variable in a relationship and behaves as the independent variable in the next relationship without affecting its previous or next paths. The hypotheses are checked through structural model stages. However, the initial data screening is done through a measurement model. The response categories ranged from 1 to 5, with 1 being a strong disagreement and 5 being a strong agreement. The questions included in the questionnaire addressed each variable (i.e., entrepreneurial finance, social capital distribution, CF, EE varieties, and entrepreneurial intention). Each variable consisted of four items. The data responses obtained through data collection were analyzed for validity and reliability through the measurement model. Results for reliability and validity can be seen in the results section.

## 3. Results

### 3.1. Demographic analysis profile of rural women of Pakistan

Table 3 shows the respondents' demographic properties. All the study's participants were females, and the age category was divided into four subcategories, with the highest percentage (47.5%) found among the 31–40 years category. Concerning marital status, most of the respondents (50.6%) were married, and most respondents (73.9%) had just primary matriculation education. Similarly, 86.85% of respondents were found to be living in a joint family.

### 3.2. Measurement model for rural women of Pakistan

The measurement model is the first step in the preliminary screening of the data collected for the study, which helps the researcher ensure that the validity and reliability among the variables are well established. Anderson & Gerbing (1988) [48] suggested assessing the measurement model's fitness before evaluating hypotheses. The present study estimated the convergent validity of the scales used through the factor loadings, Variance Inflation Factor (VIF), Cronbach's alpha, composite reliability, and the Average Variance Extracted (AVE). In Fig 3, the study's measurement model is presented, and the measurement model results are given in Table 4.

According to [49], the minimum value acceptable for the factor loadings of the items for each variable is 0.6. In the present study, the CF variable's factor loadings were above this set criteria. The minimum factor loading obtained for CF was 0.836, which meets the acceptance criteria. Similarly, the rest of the variables also showed acceptable factor loadings. The entrepreneurial finance variable had a minimum factor loading of 0.823, while the value of the entrepreneurial intention was 0.835. The Social Capital Development (SCD) variable had a minimum factor loading of 0.884, and Varieties of EE (VOE) was 0.853. Further, the collinearity diagnostic commonly used by researchers is the VIF. The VIF value should be less than 5.5 so there is no issue of multicollinearity among the study items. In this study, all items showed VIF values less than this threshold. The maximum VIF value obtained in this study was 4.67 for CF3 [50].

Further, the AVE has also been used to check the convergent validity. According to [51], an AVE value above 0.5 shows that convergent validity exists in the data. All the variables showed AVE values well above 0.5, and the minimum value was 0.743 for

Table 3. Demographics analysis profile of rural women of Pakistan.

| Demographics | Frequency | Percentage |
|---|---|---|
| **Age (years)** | | |
| 20–30 | 314 | 31.27% |
| 31–40 | 477 | 47.50% |
| 41–50 | 73 | 7.27% |
| Above 50 | 140 | 13.94% |
| **Marital Status** | | |
| Married | 508 | 50.59% |
| Unmarried | 235 | 23.40% |
| Others | 261 | 25.99% |
| **Education** | | |
| Matric | 742 | 73.90% |
| Bachelors | 262 | 26.09% |
| Masters and Studies | – | – |
| **Family System** | | |
| Joint | 872 | 86.85% |
| Nuclear | 132 | 13.14% |

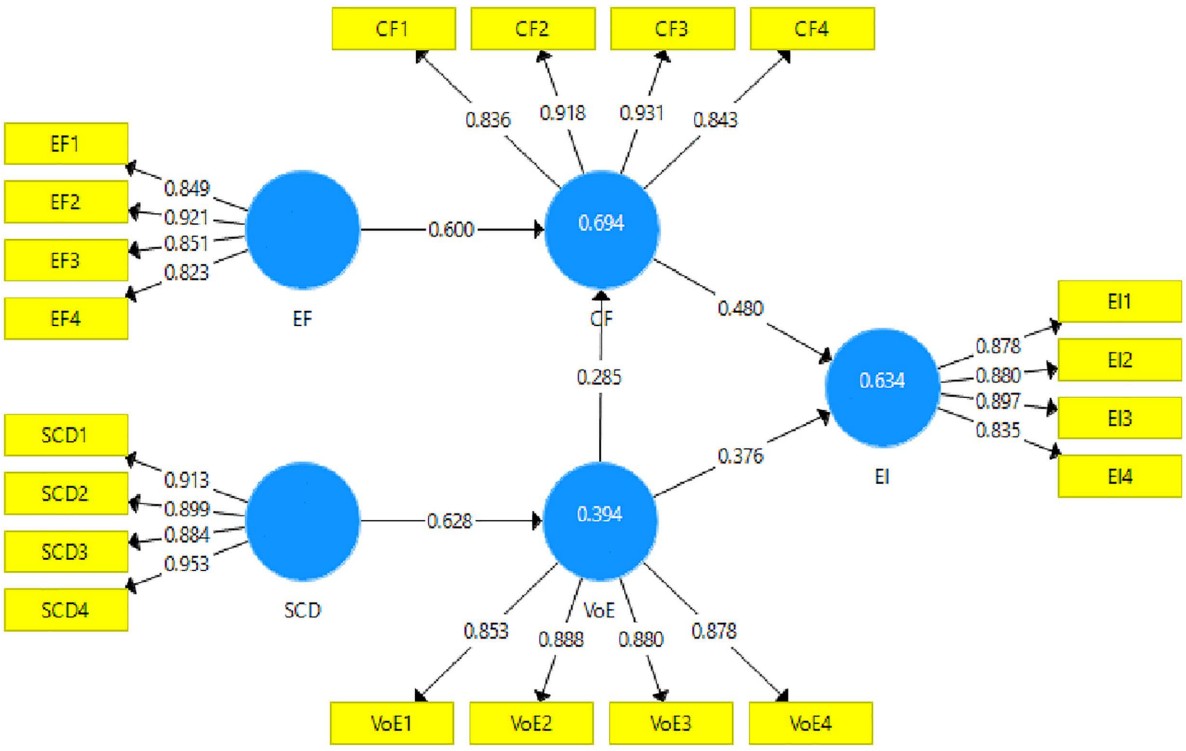

**Fig 3. Measurement model. CF** = Crowd funding, **EF** = Entrepreneurial Finance, **EI** = Entrepreneurial Intentions, **SCD** = Social Capital Distribution,
**VOE** = Varieties of Entrepreneurial Ecosystem.

the entrepreneurial intentions variable. The study also estimated the variables' reliabilities using Cronbach's alpha and composite reliability. Different studies have shown that the accepted value for reliabilities is 0.7 (i.e., the obtained values should be above this [52]. This study's minimum obtained value for Cronbach's alpha was 0.884 for entrepreneurial finance. Furthermore, the composite reliability had a minimum value of 0.913 for SCD. Thus, all the variables show convergent validities.

According to the results of structural models, the relationship between (Entrepreneurial) Finance and (Crowdfunding) is significant (about 0.6) and (Social Capital Distribution) and (Entrepreneurial Intentions) is significant (about 0.45). The mainstream tests for discriminant validity are the Fornell and Larcker Criteria and the Heterotrait–Monotrait (HTMT) ratio. Table 5 shows the results of these two tests. In the Fornell and Larcker test, the scales are valid when the highest value in each column is at the top [53]. The table shows that the highest value of each column was at the top; therefore, Table 5 indicates that the data obtained in the Fornell and Larcker table meets the acceptance criteria, showing the variables' discriminant validity. The other diagnostic test for discriminant validity absence is the HTMT ratio. Results for the HTMT ratio are given in Table 5. For HTMT ratio results to be significant, all the values in the table grid must be below 0.9 [53]. This study's results show that all the values in the table are below the set criteria, confirming the discriminant validity. The study used the coefficient of determination to explain the model's sustainability or the model fit considering the fitness to the mean regression line [54]. According to the literature, a model fit is considered good if the value obtained for R-squared is near 0.5 or 50%. The present study showed that the highest R-squared value for the CF variable was 0.694, which is a good fit. The other dependent variable, entrepreneurial intentions, showed a model fit of 0.634, indicating a good model fit. Furthermore, the VOE variable had a moderate model fit of 0.394. Overall, the variables used in this study showed a good model fit. Moreover, F-square is another indicator that checks the overall effect size on the study's

**Table 4. Measurement model assessment.**

| Variables | Factor Loadings | | VIF | Construct Reliability and Validity | | |
| --- | --- | --- | --- | --- | --- | --- |
| | | | | α | Composite Reliability | AVE |
| Crowdfunding | CF1 | 0.836 | 2.354 | | | |
| | CF2 | 0.918 | 4.115 | **0.905** | **0.934** | **0.780** |
| | CF3 | 0.931 | 4.676 | | | |
| | CF4 | 0.843 | 2.274 | | | |
| Entrepreneurial Finance | EF1 | 0.849 | 2.435 | | | |
| | EF2 | 0.921 | 3.704 | **0.884** | **0.920** | **0.743** |
| | EF3 | 0.851 | 2.439 | | | |
| | EF4 | 0.823 | 1.939 | | | |
| Entrepreneurial Intentions | EI1 | 0.878 | 3.150 | | | |
| | EI2 | 0.880 | 2.640 | **0.895** | **0.927** | **0.762** |
| | EI3 | 0.897 | 3.314 | | | |
| | EI4 | 0.835 | 2.275 | | | |
| Social Capital Distribution | SCD1 | 0.913 | 3.413 | | | |
| | SCD2 | 0.899 | 3.158 | **0.933** | **0.952** | **0.833** |
| | SCD3 | 0.884 | 3.541 | | | |
| | SCD4 | 0.953 | 6.239 | | | |
| Varieties of Entrepreneurial Ecosystem | VOE1 | 0.853 | 2.295 | | | |
| | VOE2 | 0.888 | 2.660 | **0.898** | **0.929** | **0.765** |
| | VOE3 | 0.880 | 2.783 | | | |
| | VOE4 | 0.878 | 2.708 | | | |

**Table 5. Discriminant validity.**

| Fornell–Larcker Criterion | | | | | | Heterotrait–Monotrait (HTMT) Ratio | | | | | |
| --- | --- | --- | --- | --- | --- | --- | --- | --- | --- | --- | --- |
| | CF | EF | EI | SCD | VOE | | CF | EF | EI | SCD | VOE |
| CF | 0.883 | | | | | CF | | | | | |
| EF | 0.810 | 0.862 | | | | EF | 0.907 | | | | |
| EI | 0.754 | 0.667 | 0.873 | | | EI | 0.836 | 0.749 | | | |
| SCD | 0.700 | 0.765 | 0.640 | 0.913 | | SCD | 0.761 | 0.842 | 0.700 | | |
| VOE | 0.728 | 0.738 | 0.725 | 0.628 | 0.875 | VOE | 0.807 | 0.830 | 0.806 | 0.681 | |

*CF* = *Crowdfunding*, *EF* = *Entrepreneurial Finance*, *EI* = *Entrepreneurial Intentions*, *SCD* = *Social Capital Distribution*, *VOE* = *Varieties of Entrepreneurial Ecosystem*.

relationship. According to the literature, the effect size is small if the value obtained for F-square is 0.02, medium if the value for F-square is 0.15, and large if the value of F-square is 0.35 or above [55].

This study found the largest effect size between the SCD and the VOE relationship (F-square = 0.651). Another large effect was found between the entrepreneurial finance and CF variables (F-square = 0.534). The effect size of CF and entrepreneurial intentions was also among the large effects (F-square = 0.296), along with the relationship between entrepreneurial intentions and VOE (F-square = 0.182). A medium effect size was found between the VOE and CF variables (F-square = 0.121).

### 3.3. Structural model for rural women of Pakistan

The second stage of SEM is the structural model assessment. This stage of SEM helps assess the hypotheses based on the relationships proposed in the study's hypotheses and helps assess the model using statistics, such as t-statistics, Beta

(β) values, and p-values. A pictorial representation of the structural model is given in Fig 4, and the result for the structural model can be seen in Table 6.

The structural model assessment was obtained with the help of bootstrapping at 5,000 sub-samples. T-statistics and p-values were used to conclude the hypotheses for their acceptance. The literature mentions the acceptance criteria for hypotheses is a t-statistic > 1.96 and a p-value < 0.05 [56]. Five hypotheses were developed in this study based on the literature review. Results of the hypotheses testing are given in Table 6. The first hypothesis states that CF positively impacts the entrepreneurial intentions of rural women in Pakistan. This hypothesis had a t-statistic = 10.67 and p-value < 0.05, thus accepting H1. The second hypothesis states that VOE positively impact rural women's entrepreneurial intentions in Pakistan and was found to have a t-statistic = 8.75 and p-value < 0.05, indicating the acceptance of H2. The third hypothesis of the study states that entrepreneurial finance positively impacts CF for rural women in Pakistan. The results revealed that the t-statistic for this hypothesis was 15.28 and the p-value < 0.05, showing that H3 was accepted. The study's fourth hypothesis states that VOE positively impact CF for rural women in Pakistan.

The results for this hypothesis indicate the acceptance of H4 since the t-statistic = 7.38 and the p-value < 0.05. The study's last hypothesis states that SCD for rural women in EEs in Pakistan. This hypothesis was accepted with a t-statistic = 27.27 and p-value < 0.05. In Fig 5, graphical overview of crowdfunding and varieties of entrepreneurial ecosystems for rural women in Pakistan is given.

## 4. Discussion

CF is an alternate source of finance that emerged with the development of technology and democratized financing for small businesses. The EE can potentially have a significant influence on the economy. Although CF platforms have been

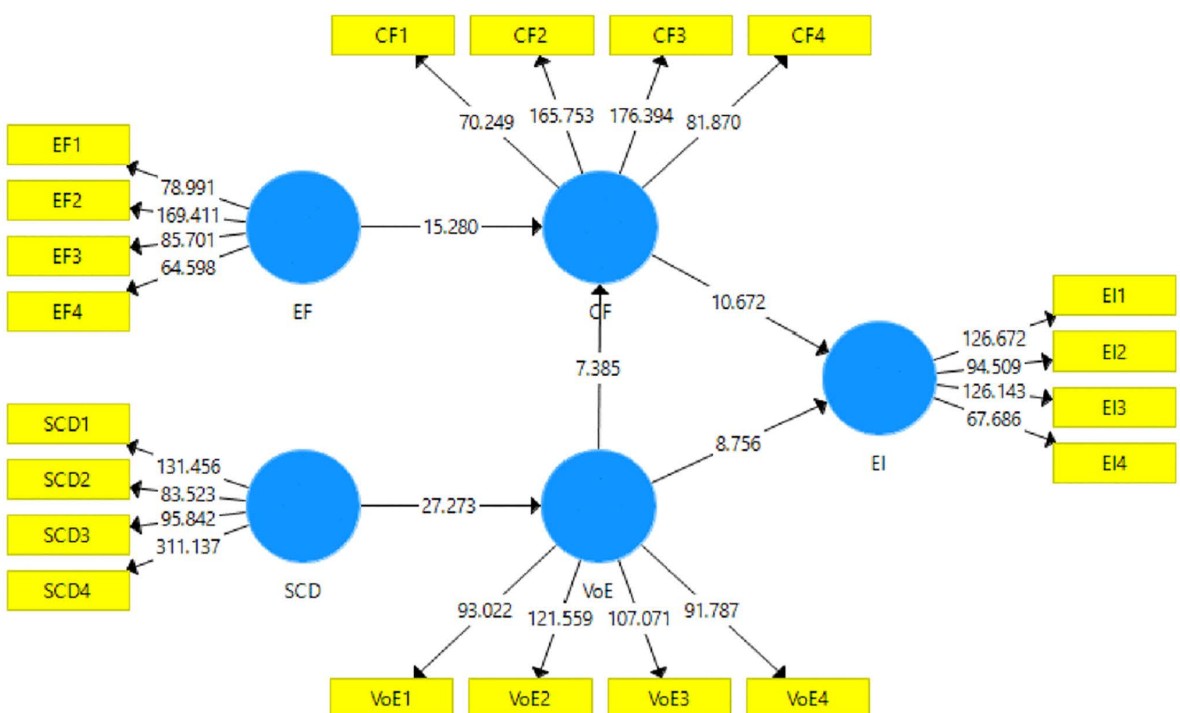

**Fig 4. Structural model of crowdfunding and varieties of entrepreneurial ecosystems for rural women in Pakistan. CF** = Crowdfunding, **EF** = Entrepreneurial Finance, **EI** = Entrepreneurial Intentions, **SCD** = Social Capital Distribution, **VOE** = Varieties of Entrepreneurial Ecosystem.

**Table 6. Direct effects of the variables.**

| Paths | H | O | M | SD | T-statistics | P-value | Results |
|---|---|---|---|---|---|---|---|
| CF -> EI | $H_1$ | 0.480 | 0.480 | 0.045 | 10.672 | **0.001** | *Accepted* |
| VOE -> EI | $H_2$ | 0.376 | 0.376 | 0.043 | 8.756 | **0.000** | *Accepted* |
| EF -> CF | $H_3$ | 0.600 | 0.599 | 0.039 | 15.280 | **0.003** | *Accepted* |
| VOE -> CF | $H_4$ | 0.285 | 0.287 | 0.039 | 7.385 | **0.000** | *Accepted* |
| SCD -> VOE | $H_5$ | 0.628 | 0.629 | 0.023 | 27.273 | **0.000** | *Accepted* |

*CF* = *Crowdfunding,* *EF* = *Entrepreneurial Finance,* *EI* = *Entrepreneurial Intentions,* *SCD* = *Social Capital Distribution,* *VOE* = *Varieties of Entrepreneurial Ecosystem*

established as an essential funding source for female entrepreneurs worldwide, research into CF's potential public policy implications has lagged. Whereas "crowdfunding" has been around since 2006, there has been little empirical study on what motivates consumer investors to support particular CF initiatives [36]. Furthermore, several researchers have suggested that CF may help reduce gender disparities in entrepreneurial funding for women [37]. The tendency for women to encourage others who are like them and to be more homophilic than males may be significant. Women are under-represented in the decision-making process for allocating funds for entrepreneurship and getting funding for running a business.

To fill these and various other gaps in the literature, we have developed some hypotheses between the independent and dependent variables based on these indicators. There is no previously available study in Pakistan in this domain; therefore, no reference was available to cite for previous results. This study is novel in the case of Pakistan and will serve as a benchmark for future research in this direction. The study investigated the effects of various EEs on rural Pakistani women's CF access. Based on the above literature research on entrepreneurship, EEs, and CF, the study has produced five hypotheses. A quantitative research design was created for this investigation to prevent biases, such as social desirability and researcher bias, from influencing the study's findings. Furthermore, the Snowball Sampling Technique was modified to gather data for the study. The Snowball sampling approach helps identify and communicate with people through social networks.

Firstly, this is one of the first studies to approach the problem through a cross-sectional analysis by quantitatively evaluating the correlations with partial least squares SEM. Prior studies have been done on different modules of years and countries compared to others. Secondly, the current study has discovered that CF for rural Pakistani women is considerably and favorably influenced by entrepreneurial finance and various EEs. Even though an EE is frequently highlighted as a crucial component of entrepreneurial success, it is still difficult to assess culture and processes reliably, especially across different areas and periods. The entrepreneurial intentions model has R2 of 0.634 which implies that this model fits well since it explains 63.4% of the variance of the dependent variable. The value of $R^2$ above 0.5 is generally regarded to be acceptable in models in social sciences, so this value is appropriate in our study [57]. Thirdly, research has added to the idea of entrepreneurship by revealing the essential and beneficial roles of VOE and CF in the entrepreneurial intentions of rural Pakistani women. According to this research, SCD significantly impacts the different EEs, influencing the entrepreneurial intentions of rural Pakistani women.

The findings indicate that the key constructs have strong relationships that include the strong effects of the construct Entrepreneurial Finance on the construct Crowdfunding (beta = 0.60). It shows that access to financial resources is one of the key factors in the success of rural women in entrepreneurship via crowdfunding. These results are aligned with the past studies, which indicate that financial access has a direct impact on the capability of the entrepreneurs to exploit crowdfunding opportunities. The beta values of other relationships, including that of Social Capital Distribution and Entrepreneurial Intentions (beta = 0.45) illustrate the role of social networks in making entrepreneurial decisions. Although these

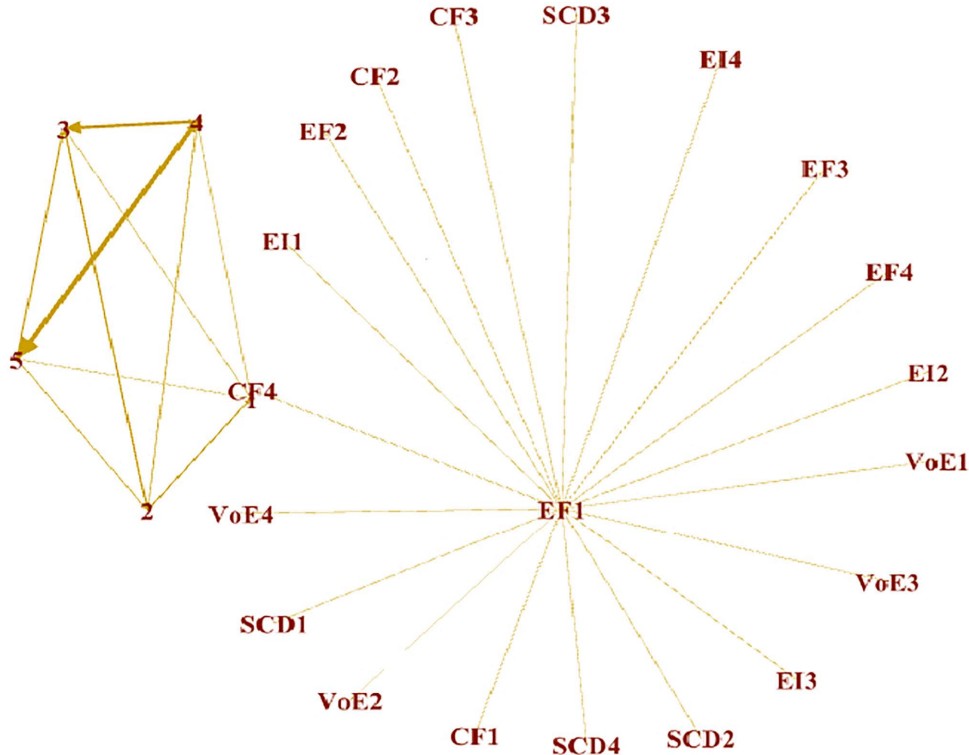

**Fig 5. Graphical Overview of CF and VOE for Rural Women in Pakistan by using Gephi 0.9.2 with Layout Fruchterman Reingold.** CF = Crowd-funding, EF = Entrepreneurial Finance, EI = Entrepreneurial Intentions, SCD = Social Capital Distribution, VOE = Varieties of Entrepreneurial Ecosystem.

statistical findings are important, the practical implications are that, policies that would raise the access to finance and the networks of social capital would have a positive effect on the entrepreneurial success of rural women.

The relationship between the independent and dependent variables for the relationships between 'Entrepreneurial Finance' and 'Crowdfunding' (f2 = 0.534) and 'Social Capital Distribution' and 'Entrepreneurial Intentions' (f2 = 0.296) hints at large and medium effects respectively. These results suggest that both financial resources and social capital are excellent predictors of entrepreneurial intentions and crowdfunding success. These findings have implications as to rural women in Pakistan: more access to entrepreneurial finance and supportive social networks can make a substantial difference in entrepreneurial processes [58].

The findings demonstrate the importance of the two concepts of 'Entrepreneurial Finance' and 'Social Capital Distribution' in influencing the entrepreneurial intentions and crowdfunding of rural women. This implies that policies that aim at enhancing avenues of accessing finance and social networks can play a decisive role in supporting rural entrepreneurship. The implications of these findings on the body of knowledge on entrepreneurship include the importance of the accessibility of finance and the availability of social networks in facilitating entrepreneurial motives. Conceptually, the findings are consistent with social capital theory, which states that the networks and relationships have huge influence on the entrepreneurial activities [59–61]. Policy Implications These findings indicate that policy interventions could be important in improving financial literacy and access to credit and social networking among rural women in Pakistan in enhancing entrepreneurship. In the example, they should think about boosting such programs as the Kamyab Jawan Program (KJP), which grants financial aid to young business owners, and involving social networking opportunities into such programs.

### 4.1. Theoretical implications

The present study makes a valuable contribution to the literature on entrepreneurship and CF. Previous research has primarily focused on panels of years and countries in comparison to others. This is among the first studies to examine the issue through a cross-sectional approach, using partial least squares SEM to quantitatively measure relationships. The study has found that entrepreneurial finance and VOE significantly and positively influence CF for rural women in Pakistan. It also advances entrepreneurship theory by highlighting the positive and significant roles of CF and VOE in shaping rural women's entrepreneurial intentions in Pakistan. Additionally, the study reveals that SCD plays a crucial part in VOE, further supporting rural women's entrepreneurial aspirations in Pakistan.

### 4.2. Practical implications

The present study offers significant practical implications for female entrepreneurs, financial institutions, investors, and the corporate sector concerning new business startups in Pakistan. It is crucial for aspiring female entrepreneurs looking to initiate new ventures or startups to develop their social capital. Such development can help them network with others who can assist in raising funds and collaborate with other rural women seeking similar opportunities. The study also enhances understanding of the rural entrepreneurship landscape and how the VOE can support them in achieving their entrepreneurial goals through a clear roadmap for CF. Consequently, deriving and interpreting the EEs becomes essential for policymakers. The participants of the study, rural Pakistani women, can aid policymakers and other stakeholders, such as investors and financial institutions, in identifying the key elements of the EEs to consider during the different stages of VOE development. This insight can help financial institutions, interested in investing in small startups and supporting rural women in Pakistan, to evaluate how effectively the EE and SCD can assist them in reaching their entrepreneurial objectives.

Although this research offers valuable insights into the importance of entrepreneurial finance and social capital for rural women entrepreneurs, there are several limitations to consider. First, snowball sampling may introduce bias because the sample is likely to reflect the social networks of the initial participants. Furthermore, while some valuable findings are provided for rural women in Pakistan, the applicability of these findings to other regions or populations is limited. Future research could be improved by employing random sampling techniques and exploring other areas to strengthen external validity. Additionally, reliance on self-reported data can lead to response bias, as participants might provide socially desirable answers.

## 5. Conclusions

Ultimately, the paper highlights the importance of improving access to finance and building social capital to empower rural women entrepreneurs in Pakistan. These findings offer valuable policy implications for policymakers aiming to create an inclusive environment for entrepreneurship. The paper explores how the allocation of entrepreneurial finance and social capital influences entrepreneurial intentions and the success of crowdfunding among rural women in Pakistan. The results underline the significance of financial accessibility and social networks in fostering entrepreneurship, impacting policy development.

CF is a novel approach in developing countries for promoting and financing the expansion of businesses, entrepreneurs, and ecosystems in general. In this study, a few determining factors were studied and found important for the rural entrepreneurs to help them in CF and small businesses. The SEM has also helped determine the positive and significant contributions provided by various EEs in CF, and these two have been demonstrated to predict the entrepreneurial intentions of rural Pakistani women strongly. The study also gives a better insight into the scenario of rural entrepreneurship for rural Pakistani women and how the VOE can help them achieve their entrepreneurial plans through a solid CF roadmap. The results found significant and positive relationships among those factors. The SEM has further helped in finding the positive and significant contribution of VOE in CF, and these two have been found to predict the entrepreneurial intentions

of the rural women of Pakistan. Therefore, it will serve as a benchmark for future studies and research on such issues. Data inventory, methodological approach, and unique results make this study novel in filling the research gap. The current study represents a new contribution as it reveals the evidence of the direct and indirect channels through which social capital and entrepreneurial finance affect rural women in accessing crowdfunding opportunities. The findings address a significant gap in the women entrepreneurship literature of South Asia and provide a practical model that can be used to help rural women entrepreneurs in Pakistan.

This study contributes as a way forward to the literature and to the policymakers. Furthermore, the VOE is an emerging concept in the entrepreneurship literature. Hence, the generalizability of the results open opportunities for other researchers interested in this field to show interesting findings. This is also important to understand the VOE in different settings and countries because EEs may get weaker over time rather than strengthening or perhaps face a variety of compressed economic developments. This possibility of varying trajectories in the life cycles can modify the interpretation of the results for other countries. Therefore, more research is needed to assess the circumstances in different territories. Future studies can attempt further operationalization of the present framework proposed by modifying it according to the contextual EE, which can be better used.

## Supporting information

**S1 Table. Questionnaire.**
(DOCX)

## Author contributions

**Conceptualization:** Zhikang Lyu.

**Data curation:** Zhikang Lyu.

**Formal analysis:** Natasha Murtaza.

**Investigation:** Zhikang Lyu.

**Methodology:** Zhikang Lyu.

**Project administration:** Zhikang Lyu.

**Software:** Natasha Murtaza.

**Supervision:** Zhikang Lyu.

**Validation:** Zhikang Lyu.

**Visualization:** Natasha Murtaza.

**Writing – original draft:** Zhikang Lyu.

**Writing – review & editing:** Natasha Murtaza.

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
