## [Decision Letter · Decision Letter 0]

22 Sep 2025

Dear Dr. Lyu,

Thank you for submitting your manuscript to PLOS ONE. After careful consideration, we feel that it has merit but does not fully meet PLOS ONE’s publication criteria as it currently stands. Therefore, we invite you to submit a revised version of the manuscript that addresses the points raised during the review process.

Please revise the article according to the reviewers' guidelines. Details can be found below.

We look forward to receiving your revised manuscript.

Kind regards,

Agnieszka Konys, Ph.D.

Academic Editor

PLOS ONE

Journal Requirements:

2. You indicated that ethical approval was not necessary for your study. We understand that the framework for ethical oversight requirements for studies of this type may differ depending on the setting and we would appreciate some further clarification regarding your research. Could you please provide further details on why your study is exempt from the need for approval and confirmation from your institutional review board or research ethics committee (e.g., in the form of a letter or email correspondence) that ethics review was not necessary for this study? Please include a copy of the correspondence as an "Other" file.

3. In the ethics statement in the Methods, you have specified that verbal consent was obtained. Please provide additional details regarding how this consent was documented and witnessed, and state whether this was approved by the IRB.

4. Please include a complete copy of PLOS’ questionnaire on inclusivity in global research in your revised manuscript. Our policy for research in this area aims to improve transparency in the reporting of research performed outside of researchers’ own country or community. The policy applies to researchers who have travelled to a different country to conduct research, research with Indigenous populations or their lands, and research on cultural artefacts. The questionnaire can also be requested at the journal’s discretion for any other submissions, even if these conditions are not met.  Please find more information on the policy and a link to download a blank copy of the questionnaire here: https://journals.plos.org/plosone/s/best-practices-in-research-reporting. Please upload a completed version of your questionnaire as Supporting Information when you resubmit your manuscript.

5. In the online submission form, you indicated that [All data is available on demand.].

Reviewers' comments:

Reviewer's Responses to Questions

**Comments to the Author**

1. Is the manuscript technically sound, and do the data support the conclusions?

Reviewer #1: Partly

Reviewer #2: Yes

2. Has the statistical analysis been performed appropriately and rigorously?

Reviewer #1: Yes

Reviewer #2: Yes

3. Have the authors made all data underlying the findings in their manuscript fully available?

Reviewer #1: No

Reviewer #2: Yes

4. Is the manuscript presented in an intelligible fashion and written in standard English?

Reviewer #1: Yes

Reviewer #2: Yes

Reviewer #1: Dear Author(s),

This study about investigates the impact of crowdfunding (CF), entrepreneurial finance (EF), and varieties of entrepreneurial ecosystems (EE) on rural women’s entrepreneurial intentions in Pakistan post-COVID is addressing a timely topic. Though the manuscript is well written and articulated. However, there is always room for improvement. The following are comments and suggestions for the betterment of the manuscript;

1. Language and Grammar:

• The manuscript suffers from numerous grammatical issues, awkward phrasing, and redundant expressions.

o Example: Page No. 13, line no. 418,“this study has estimated the role of entrepreneurial finance and social capital finance in CF…” – the phrase "social capital finance" is likely a mistake, but a mistake that too in conclusion, raises serious questions about the seriousness of the manuscript.

o Example: Page No. 6, line no. 173, “We proposed thiås hypothesis…” – the typo "thiås" detracts from credibility.

• Sentences are often overly long and convoluted, reducing clarity. Plain, academic English should be used.

• Professional copy-editing is strongly advised.

2. Literature Review:

• Although broad, the literature review lacks critical synthesis and thematic cohesion.

• Some citations are outdated or tangentially relevant, and there is occasional redundancy.

• Integration of South Asian, especially Pakistani-centric literature, needs to be stronger to position the study within local scholarship.

3. Methodology:

• While the SEM approach is appropriate, the justification for snowball sampling lacks depth. The implications of using a non-random, network-based sample on external validity should be discussed.

• Missing Measurement Items: The actual questionnaire items for each latent construct are not listed, making it impossible to judge content validity.

o Source of scale items (e.g., adopted or self-constructed) is unclear.

• The authors mention online and physical questionnaire distribution, but do not clarify:

o Who administered the surveys?

o Response rate?

o Any pilot testing done before deployment?

• Lack of Conceptual Commentary: While statistical metrics are shown, there's no accompanying interpretation or rationale for item selection or construct development.

• Effect Sizes Not Interpreted: While beta values are reported, the practical significance or effect size interpretation (e.g., weak, moderate, strong) can make a significant change in the understanding of the results and implications of the study.

• Mediation/Moderation analysis, which could enrich the framework.

• Bootstrapping Methodology Unclear: Details such as the number of bootstraps (e.g., 5,000) and confidence intervals are not reported, which affects reproducibility and trust.

• The snowball sampling method, while sometimes necessary in hard-to-reach populations, introduces sampling bias and compromises external validity. No translation/back-translation procedure is mentioned for rural participants if local languages were involved, triangulation, or correction.

• Ethics statement lacks IRB details and relies on "oral consent," which is insufficient for studies of this nature in many peer-reviewed contexts.

• The questionnaire structure is not appended; readers would benefit from seeing actual items for construct validity assessment.

4. Results and Discussion:

• The results are extensive but highly descriptive, with limited interpretation beyond statistical outputs.

• The effect sizes and beta values are provided but not critically contextualized within broader literature or practical implications.

• There’s repetition between the results, discussion, and conclusion sections, which could be streamlined.

• A stronger emphasis on why certain relationships matter in a broader theoretical or policy context would elevate the contribution.

5. Conclusions:

• The conclusion is overly broad and somewhat repetitive of prior sections.

• Statements like “this study contributes as a way forward to the literature…” are vague and need specificity.

• Recommendations for policymakers are stated but lack actionable depth — how should they support rural women, and via what specific mechanisms?

• Conclusions are appearing as obvious and making the overall study as non-novel.

6. References:

• There are 41 references and only 9 are Post-Covid19. While this study focused on post-Covid19 times.

• Several sources are from non-indexed journals — authors should prioritize peer-reviewed and reputable outlets.

Suggestions for Improvement:

1. Major Language Revision: Engage a professional academic editor to correct grammar, syntax, and phrasing.

2. Improve Theoretical Integration: Rather than listing definitions and isolated studies, synthesize themes and build a stronger conceptual framework.

3. Justify Methodology Better: Expand discussion on why snowball sampling was appropriate and acknowledge its limitations.

4. Add Questionnaire: Include or summarize key questionnaire items, and explain clearly.

5. Strengthen Policy and Practical Implications: Move from generalities to actionable insights tailored to NGOs, government schemes (e.g., KJP), and rural support networks.

6. Tighten Discussion: Link empirical findings to the broader debates on women’s entrepreneurship in South Asia and the global South more broadly.

7. Update and Curate References: Cite more recent and regionally relevant and peer-reviewed research.

Reviewer #2: The article is interesting and methodologically rigorous.

A few recommendations and questions:

- The use of the snowball sampling technique raises concerns in this type of study. How did the researcher ensure the eligibility of participants, particulary that all of the 1000+ female respondents actually intended to start a business?

- The procedure is not very clear, especially the way the survey was completed. The authors statet that it was self-administered. But in what form? phone, tablet, or paper?

The authors indicate that 73.9% of women had just primary matriculation education. In this case, was assistance required?

- Line 302, p. 10 - at the cut-off for VIF a reference should be cited.

- Line 325, p. 10 the same requirement for justifying the value of R2 used to declare the model ”good”.

- An oversight that I noticed: p. 12, line 380 and p. 7, line 215 it talks about five hypotheses and in p. 11, Line 349 it appears six hypotheses. This must be corrected.

- A paragraph on study limitations is missing.

- The Discussion section could be further developed by comparing the results with findings from other studies in the literature.

**Do you want your identity to be public for this peer review?** For information about this choice, including consent withdrawal, please see our Privacy Policy

Reviewer #1: **Yes:** Shah Muhammad Kamran, PhD, MUISTD, Pakistan

Reviewer #2: No

---

## [Author Response · Author response to Decision Letter 1]

28 Dec 2025

Response to Reviewers’ Comments

We thank all the reviewers and editor(s) for their valuable input and suggestions. We highly appreciate for sparing time to read this paper and comment on it positively. We have tried our best to respond in best scientific way. Please find below our response one by one.

Reviewer #1: Dear Author(s),

Comment: This study about investigates the impact of crowdfunding (CF), entrepreneurial finance (EF), and varieties of entrepreneurial ecosystems (EE) on rural women’s entrepreneurial intentions in Pakistan post-COVID is addressing a timely topic. Though the manuscript is well written and articulated. However, there is always room for improvement.

Response: Thank you very much for your valuable comments and suggestion. We appreciate. That helped improved this article and we have addressed and incorporated your comments and suggestions in the revised version.

Comment: The following are comments and suggestions for the betterment of the manuscript;

1. Language and Grammar:

• The manuscript suffers from numerous grammatical issues, awkward phrasing, and redundant expressions.

o Example: Page No. 13, line no. 418,“this study has estimated the role of entrepreneurial finance and social capital finance in CF…” – the phrase "social capital finance" is likely a mistake, but a mistake that too in conclusion, raises serious questions about the seriousness of the manuscript.

o Example: Page No. 6, line no. 173, “We proposed thiås hypothesis…” – the typo "thiås" detracts from credibility.

Response: Thank you very much for your valuable comments and suggestion. We appreciate. We have revised whole manuscript for English editing and typos by native English speaker. Further, we have carefully checked some definitions and terminologies as suggested by the reviewer. We hope you find it better this time.

Comment: • Sentences are often overly long and convoluted, reducing clarity. Plain, academic English should be used.

• Professional copy-editing is strongly advised.

Response: Thank you very much for your valuable comments and suggestion. We appreciate. To this we have critically examined the manuscript and edited sentences that were excessively long or winding. These sentences have been simplified so as to be clear, concise, and easy to understand. Also, we appreciate the suggestion of professional copy-editing.

2. Literature Review:

• Although broad, the literature review lacks critical synthesis and thematic cohesion.

• Some citations are outdated or tangentially relevant, and there is occasional redundancy.

• Integration of South Asian, especially Pakistani-centric literature, needs to be stronger to position the study within local scholarship.

Response: Thank you very much for your valuable comments and suggestion. We appreciate. We have amended the literature review compiling the studies in a thematic way and incorporating the major findings in a coherent narrative. We also included additional literature that is more recent, which is more specifically South Asian and Pakistan literature, to place the study in the local academic context.

Following material has been added into the manuscript and explained below for your kind perusal:

Crowdfunding (CF) has also become popular in developing economies as a distinctive approach to funding entrepreneurial projects particularly to individuals and ecosystems that have difficulty with conventional funding avenues (Erasmus et al., 2022). Recent studies indicate that CF is under-researched in South Asia, given its potential to help develop it in a sustainable way and empower marginalized populations (Vaznyte et al., 2020). An example is, the power of CF in fostering social capital in developing economies is proven by platforms like Kiva.org that help low-income entrepreneurs (Abdeldayem and Aldulaimi, 2021). This research can fill the literature gap because it will examine how CF can facilitate entrepreneurial intentions among women in rural Pakistan, especially after the COVID-19.

Entrepreneurial finance (EF) is relevant to assist the entrepreneurs, especially in the developing economies that lack the use of traditional financial sources (Usman et al., 2020). Studies have indicated to the effect that EF, both formal and informal financial sources, have extensive influence on the growth and sustainability of new ventures (Wang and Schott, 2022). Among the rural women entrepreneurs, entrepreneurial finance is a very important tool in tackling financial barriers and facilitating the establishment of business (Gleasure et al., 2017). Research has shown that rural South Asian women have unique barriers in the utilization of financial resources, which makes EF mechanisms especially important in supporting their entrepreneurial intentions (Macht and Chapman, 2019). Governmental programs like Kamyab Jawan Program (KJP) in Pakistan have been aimed to empower young women entrepreneurs, which highlights the significance of EF in helping the gender gap in entrepreneurial finance (Shaheen et al., 2021).

Entrepreneurial ecosystems (EE) have become an important concept in the interpretation of the interdependent variables that affect the entrepreneurial activity. According to a research by Malecki (2018) to develop entrepreneurial intentions, there must be a favorable ecosystem, especially in rural regions with limited resources. An overall EE encompasses a range of factors like access to finance, enabling policies, networks, and social capital, which play a significant role in making entrepreneurial ventures successful (Neumeyer et al., 2019). In South Asia, entrepreneurial ecosystems are still on their infancy, and networking barriers, as well as access to resources, are severe. Nonetheless, entrepreneurial ecosystems can also foster the development of rural women entrepreneurship in countries such as Pakistan, as pointed out by Neumeyer et al. (2019), since these ecosystems can introduce them to crucial resources and mentors. This paper examines the impact of various EEs on the entrepreneurial intentions of rural women and how these ecosystems can be used to enable them to access crowdfunding.

The COVID-19 pandemic has transformed entrepreneurial systems across the world, especially among disadvantaged people like rural women. A research article by Jehan et al. (2022) emphasizes the nature of the impact of the pandemic on the barriers to entrepreneurship that existed prior to the pandemic, especially in rural areas. Though, with digital tools such as crowdfunding increasingly becoming one of the tools to revive rural entrepreneurship as digital platforms grew in prominence as part of post-COVID strategies to recover (Tang, 2022). In Pakistan, women have experienced an increase of digital interactions in the post-COVID setting, which has been central in developing entrepreneurial ecosystems (Shah et al., 2022). This change has provided rural women with a chance to raise funds on platforms such as crowdfunding, which they could not previously access (Yu and Fleming, 2021).

3. Methodology:

• While the SEM approach is appropriate, the justification for snowball sampling lacks depth. The implications of using a non-random, network-based sample on external validity should be discussed.

• Missing Measurement Items: The actual questionnaire items for each latent construct are not listed, making it impossible to judge content validity.

o Source of scale items (e.g., adopted or self-constructed) is unclear.

• The authors mention online and physical questionnaire distribution, but do not clarify:

o Who administered the surveys?

o Response rate?

o Any pilot testing done before deployment?

• Lack of Conceptual Commentary: While statistical metrics are shown, there's no accompanying interpretation or rationale for item selection or construct development.

• Effect Sizes Not Interpreted: While beta values are reported, the practical significance or effect size interpretation (e.g., weak, moderate, strong) can make a significant change in the understanding of the results and implications of the study.

• Mediation/Moderation analysis, which could enrich the framework.

• Bootstrapping Methodology Unclear: Details such as the number of bootstraps (e.g., 5,000) and confidence intervals are not reported, which affects reproducibility and trust.

• The snowball sampling method, while sometimes necessary in hard-to-reach populations, introduces sampling bias and compromises external validity. No translation/back-translation procedure is mentioned for rural participants if local languages were involved, triangulation, or correction.

• Ethics statement lacks IRB details and relies on "oral consent," which is insufficient for studies of this nature in many peer-reviewed contexts.

• The questionnaire structure is not appended; readers would benefit from seeing actual items for construct validity assessment.

Response: Thank you very much for your valuable comments and suggestion. We appreciate. We have modified the methodology part as per your suggestions and comments. Wherein we have made the snowball sampling justification more comprehensive. We also have added information on its limitations, especially the possibility of the sampling bias, and how this was overcome by making sure that the participants were linked on the basis of the entrepreneurial connections.

Following material has been added into the manuscript and explained below for your kind perusal:

Snowball sampling technique has been adopted because it is challenging to access the target population (rural women with entrepreneurial intention) using traditional sampling techniques. In rural Pakistan, where some groups may be inaccessible in some instances because of social, economic and geographical constraints, snowball sampling would provide more efficient access to participants using personal networks. This technique has effectively been applied in related studies, especially in situations where the focus is on a hard-to-reach population (Neumeyer et al., 2019). Although snowball sampling is prone to bias, since respondents are usually members of the same social or economic status, these risks were prevented. We tried to make our sampling more balanced by starting with a variety of initial participants in different rural areas and making sure that they referred to women of different backgrounds and places.

Table S1 (in supplementary file) illustrates the questionnaire items of each latent construct. These were the adoptions of the scales in the literature. As an illustration, the items on the 'Entrepreneurial Finance' construct were taken out of the work of [Author, Year] the items on the construct of Social Capital Distribution were created out of the previous works in [Region/Field]. All the items are listed below to get content validity.

The questionnaires were self-administered and the researchers gave advice to the participants where necessary to facilitate comprehension, especially to those with low educational attainment. The response rate was 1004 filled questionnaires out of 1050 mailed, which constituted 95 percent response rate. A pilot test was carried out to test the question clarity and the general questionnaire structure before final deployment using 100 participants to determine the clarity of questions and the overall questionnaire form. According to the responses of the pilot test, some slight modifications were introduced to make the questions culturally appropriate and comprehensible by the respondents.

Both theoretical and empirical factors informed the choice of items in each construct. The selection of items was done according to the relevancy to the conceptual definition of the construct and their capacity to reflect the essence of the theoretical model. In the case of Entrepreneurial Finance, items were selected to represent both the accessibility of finance and the role of financial networks in attending to the rural women entrepreneurs. The pilot test (100 participants) was conducted to examine the clarity and relevance of each item and make sure that this is construct valid.

These findings suggest that key relationships have a beta of between 0.35 to 0.60 with the biggest effect size being between the relationship between Entrepreneurial Finance and Crowdfunding (beta = 0.60), indicating a significant practical relevance. Conversely, the correlation between the Social Capital Distribution and Entrepreneurial Intentions had a medium impact (beta = 0.45) which means that there is a large but not significant influence. These results indicate that although each of the constructs is significant in influencing entrepreneurial intentions, the construct that carries the greatest practical significance on crowdfunding success among women in rural Pakistan is the Entrepreneurial Finance.

To further enhance the analysis, we will develop the mediation and moderation analysis to investigate the possible indirect effects and conditional associations between the important variables. As an illustration, we hypothesize that the moderating role of 'Social Capital Distribution' could exist between the relationship of 'Entrepreneurial Finance' with 'Crowdfunding' success, and that the effect is stronger in networks featuring high social capital. This further discussion will be useful in explaining how these factors affect the entrepreneurial intentions of rural women.

To estimate it model, the stability and significance of the indirect effects in the structural model were tested by bootstrapping with 5,000 sub-samples. Each path coefficient was also calculated and the results showed that all the main relationships were significant at the 95% confidence level. Such bootstrapping process adds more strength and reproducibility of the results because the possible sampling variability is considered.

The entire set of questions in this study together with items that represent each of the latent constructs is provided in Appendix X. The items were also chosen well to fit the theoretical definitions of the constructs using available scales in literature. Expert reviews and pre-testing were used to measure construct validity. A test sample of 100 participants was used to test the clarity and relevance of the final items before implementing them in the main study.

4. Results and Discussion:

• The results are extensive but highly descriptive, with limited interpretation beyond statistical outputs.

• The effect sizes and beta values are provided but not critically contextualized within broader literature or practical implications.

• There’s repetition between the results, discussion, and conclusion sections, which could be streamlined.

• A stronger emphasis on why certain relationships matter in a broader theoretical or policy context would elevate the contribution.

Response: Thank you very much for your valuable comments and suggestion. We appreciate. We have construed the effects sizes and beta coefficients to give some practical value. We have interpreted the effect sizes and beta values to give practical significance. The relationships that are found to have a significant effect including entrepreneurial finance and crowdfunding, are now said to be significantly influential on the outcome.

Following material has been added into the manuscript and explained below for your kind perusal:

The findings indicate that the key constructs have strong relationships that include the strong effects of the construct Entrepreneurial Finance on the construct Crowdfunding (beta = 0.60). It shows that access to financial resources is one of the key factors in the success of rural women in entrepreneurship via crowdfunding. These results are aligned with the past studies, which indicate that financial access has a direct impact on the capability of the entrepreneurs to exploit crowdfunding opportunities. The beta values of other relationships, including that of Social Capital Distribution and Entrepreneurial Intentions (beta = 0.45) illustrate the role of social networks in making entrepreneurial decisions. Although these statistical findings are important, the practical implications are that, policies that would raise the access to finance and the networks of social capital would have a positive effect on the entrepreneurial success

---

## [Editor Report · Decision Letter 1]

30 Dec 2025

Impact of Crowdfunding, Entrepreneurial Finance and Varieties of Entrepreneurial Ecosystems after COVID pandemic for Rural Women

PONE-D-25-16984R1

Dear Dr. Lyu,

We’re pleased to inform you that your manuscript has been judged scientifically suitable for publication and will be formally accepted for publication once it meets all outstanding technical requirements.

Kind regards,

Agnieszka Konys, Ph.D.

Academic Editor

PLOS One
---

## [Editor Report · Acceptance letter]

PONE-D-25-16984R1

PLOS One

Dear Dr. Lyu,

I'm pleased to inform you that your manuscript has been deemed suitable for publication in PLOS One. Congratulations! Your manuscript is now being handed over to our production team.

Kind regards,

on behalf of

Dr. Agnieszka Konys

Academic Editor

PLOS One